SOFTWARE

# OmniSegger: A time-lapse image analysis pipeline for bacterial cells

**Teresa W. Lo**[iD][1], **Kevin J. Cutler**[1], **H. James Choi**[1], **Paul A. Wiggins**[iD][1,2,3]*

**1** Department of Physics, University of Washington, Seattle, Washington, United States of America, **2** Department of Bioengineering, University of Washington, Seattle, Washington, United States of America, **3** Department of Microbiology, University of Washington, Seattle, Washington, United States of America

* pwiggins@uw.edu

## Abstract

Time-lapse microscopy is a powerful tool to study the biology of bacterial cells. The development of pipelines that facilitate the automated analysis of these datasets is a long-standing goal of the field. In this paper, we describe the *OmniSegger* pipeline developed as an open-source, modular, and holistic suite of algorithms whose input is raw microscopy images and whose output is a wide range of quantitative cellular analyses, including dynamical cell cytometry data and cellular visualizations. The updated version described in this paper introduces two principal refinements: (i) robustness to cell morphologies and (ii) support for a range of common imaging modalities. To demonstrate robustness to cell morphology, we present an analysis of the proliferation dynamics of *Escherchia coli* treated with a drug that induces filamentation. To demonstrate extended support for new image modalities, we analyze cells imaged by five distinct modalities: phase-contrast, two brightfield modalities, and cytoplasmic and membrane fluorescence. Together, this pipeline should greatly increase the scope of tractable analyses for bacterial microscopists.

**Data availability statement:** Phase-contrast and fluorescence models. The Omnipose phase-contrast model and fluorescence model

## Author summary

A new generation of machine learning algorithms is pushing the boundaries of what is possible with automated analysis of microscopy images. In this paper, we describe a new tool, OmniSegger, which we recently developed to automate time-lapse image analysis. The tool was designed to solve two specific challenges: the robust analysis of bacterial cells with unusual morphologies and the use of a range of imaging modalities. Both of these challenges emerged organically in our own work and provides context for the pipeline development. Importantly, we find that the new package facilitates a previously untractable analysis of essential gene knockouts. These mutations lead to dramatic morphological changes before growth arrest and to very poor analysis performance from existing packages. In contrast, the OmniSegger pipeline can analyze these datasets

are included with Omnipose (bact_phase_omni and bact_fluor_omni):
https://doi.org/10.1038/s41592-022-01639-4.
Brightfield model. The Omnipose brightfield model is available on Zenodo at
https://doi.org/10.5281/zenodo.14225611
under the CC-BY-NC 4.0 license. Brightfield datasets. Bacterial brightfield and fluorescence image sets of *Escherichia coli* and *Burkholderia thailandensis* were generated in this study. Additional *E. coli* and *Staphylococcus aureus* brightfield images were sourced from a selection of the DeepBacs datasets which is available under the CC-BY 4.0 license. The brightfield images and ground truth masks used to train the brightfield Omnipose model are available on Zenodo at
https://doi.org/10.5281/zenodo.14225852
under the CC-BY-NC 4.0 license. Analysis datasets. Image and OmniSegger mask files of the data presented in this paper are available on Zenodo at
https://doi.org/10.5281/zenodo.14225892
under the CC-BY-NC 4.0 license. Code availability. OmniSegger package. The code for OmniSegger is available on GitHub:
https://github.com/tlo-bot/omnisegger/.

**Funding:** H.J.C., T.W.L., and P.A.W. were supported by National Institutes of Health grant R01-GM128191 and National Science Foundation grant GR046955. K.J.C. was supported by the Molecular Biophysics Training Program (National Institutes of Health grant T32GM008268). The funders did not play a role in the study design, data collection and analysis, decision to publish, or preparation of the manuscript.

**Competing interests:** The authors have declared that no competing interests exist.

and does not require the fine-tuning of a custom segmentation model. The pipeline is powered by a new segmentation algorithm, Omnipose, which we recently described elsewhere as a general-purpose cell segmentation algorithm for the analysis of single images. The current paper describes a complete time-lapse image analysis pipeline suitable for bacterial cell biology.

## 1. Introduction

Time-lapse microscopy is a powerful tool for understanding the structure and function of bacterial systems. In a single field of view, it is possible to capture the dynamics of hundreds to thousands of cells simultaneously [1]. The quantitative analysis of this image data can involve a wide range of cell cytometry, defined as the determination of morphological characteristics. Although these algorithms have a long history [2], the advent of machine-learning-based approaches has greatly facilitated the development of ever more robust and precise tools [3].

Ten years ago, we developed *SuperSegger*, an image analysis pipeline for the analysis of time-dependent fluorescence localization in bacterial cells [4]. Its development was motivated by the challenge of performing a proteome-wide time-lapse analysis of localization of nearly all proteins with non-diffuse localization in *Escherichia coli* [1,5]. However, since releasing the original SuperSegger analysis pipeline, we have discovered numerous scenarios in which the package fails to produce acceptable results.

In this paper, we highlight two types of analyses that will serve as the motivation for the development of a new package: diverse cell morphologies and the use of alternative image modalities. We will demonstrate a new image analysis pipeline, OmniSegger, which combines a new segmentation package we recently developed, Omnipose [3], with the existing analysis pipeline, SuperSegger [4]. We demonstrate that OmniSegger has superior performance to existing pipelines, both in the context of imaging standard time-lapse data, as well as vastly superior performance in the context of the morphology and modality challenges.

## 2. Design and implementation

**OmniSegger pipeline overview.** The overall goal of the OmniSegger pipeline is to provide an open-source, modular, and holistic suite of algorithms whose input is raw multichannel microscopy images and whose output is a comprehensive range of quantitative cellular analyses presented in a user-friendly format that does not require coding expertise. However, we have made the algorithms modular to facilitate the use of individual modules by users who wish to code their own custom pipelines. A schematic representing the OmniSegger pipeline and a gallery of images and analyses generated using OmniSegger is shown in Fig 1.

*Image data input.* The algorithm is designed to process image data with multiple x-y positions, time points, and fluorescence image channels. The algorithm inputs data from image files following a naming scheme used by the Nikon Elements image export tool; in addition, we include a utility that facilitates the conversion of data into this format.

*Image registration.* Although many modern microscopes use encoded stages, samples are still subject to undesired motion on the micron scale, called *drift* or *jitter*. To minimize the effect of drift, we apply an optional image registration step, on a user-selected channel, to align successive images with sub-pixel resolution using a cross-correlation registration algorithm [7]. We find this step greatly improves both the time-lapse movies generated for visualization, and the robustness of the cell linking analysis.

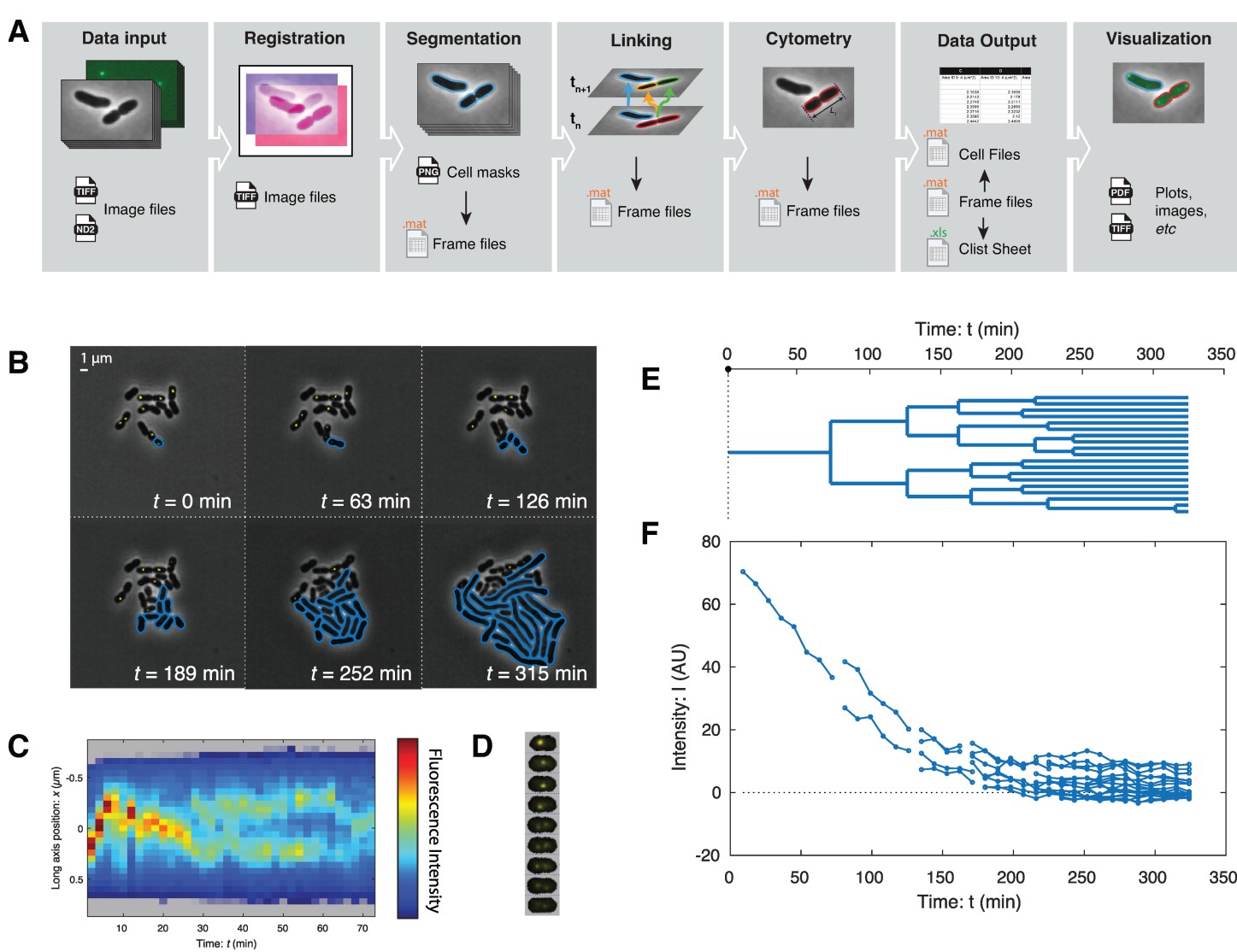

**Fig 1. OmniSegger: A microscopy analysis pipeline featuring a wide range of tools for quantitative analysis. Panel A: OmniSegger pipeline schematic. Data input:** Multi-dimensional image data is loaded from image files. **Registration:** Images are registered to remove stage drift. **Segmentation:** For each position and time-point, the first channel image is segmented to generate the cell masks, which are saved as PNG files. These masks are then incorporated into a frame file, which is a composite data file containing all image information (all channels and cell masks). The cell masks PNG is editable. **Linking:** Cell masks from successive time points are then linked to form cell trajectories, including cell division. The masks and links are corrected for temporal consistency and subsequently saved into the frame files. **Cytometry:** Cell cytometry information for each cell is computed from the image information in the frame files. **Data output:** The output data is sliced into three different output formats: The *frame files* contain all information, including images, grouped by frame (*i.e.* all cells per time-point and x-y position). The *cell files* contain all information, including images, grouped by cell (*i.e.* all time-points per cell). The *clist file* contains all cytometry information (no image information) grouped per x-y position. **Visualization:** The package also contains numerous visualization tools which use the output data to generate figures, images, and plots. **Panels B–F: OmniSegger visualization gallery.** The pipeline includes numerous tools to visualize cellular growth dynamics. **Panel B: Frame mosaics show proliferation at a multi-cellular scale.** OmniSegger can generate multi-channel composite images and supports the use of vector (rather than raster) cell outlines for improved figures. This panel highlights the ability of OmniSegger to analyze cells with a filamentous phenotype. **Panel C: Kymographs show the intracellular dynamics.** The panel shows the long-axis localization of the replisome (labeled by YPet-DnaN) using false color. **Panel D: Cell towers show intracellular dynamics in 2D.** The panel shows the 2D localization of the replisome (labeled by YPet-DnaN). **Panel E: Lineage trees show cell proliferation from a single progenitor. Panel F: Automated cell cytometry facilitates quantitative analysis.** OmniSegger automatically generates over 100 cellular descriptors, including average fluorescence intensity. In this experiment, the targeted protein YPet-DnaN is depleted by cell-proliferation-induced dilution [6].

*Cell segmentation.* The automated detection of the regions in an image corresponding to individual cells is called *instance segmentation* (segmentation) [8]. In the pipeline, the segmentation algorithm works on the first channel of the multi-channel images, and now supports the segmentation of multiple imaging modalities. OmniSegger implements our Omnipose package [3], introducing significant improvements to segmentation performance in comparison to our original release, SuperSegger. The Omnipose algorithm uses a flexible machine-learning-based approach that is well suited to both the problem of segmenting unusual cell morphologies and the segmentation of alternative imaging modalities. The updated algorithm outputs cell masks as PNG files that can be manually edited if segmentation errors arise.

*Expanded masks.* The original SuperSegger package had an important limitation in defining cell boundaries: It required a pixel gap between neighboring cells to define the cell masks; however, when cells grow in microcolonies, they are typically in contact. For example, for a pixel size of 100 nm and cell area of 3 $\mu$m$^2$, if we assume boundaries on the cell edges are receded by at least one pixel on each edge, the mask will fail to capture about 10% of the cell, even in the absence of segmentation errors. SuperSegger's original segmentation predicted binary masks for each cell (semantic segmentation); Omnipose predicts uniquely labeled cell masks (instance segmentation). Therefore, masks created in the updated pipeline can be in contact, eliminating this boundary artifact. The improved boundaries defined by the cell masks enable more accurate and consistent measurements of cell properties on the subcellular scale (see Fig 2).

*Linking/tracking.* The segmentation algorithm defines cell regions at each time point of a time-lapse experiment; however, to study cellular dynamics, the regions in successive time points must be linked to form cell trajectories. From the labeled mask image, each label assigns a region number to a cell, which is determined on a per-frame basis–region numbers are not persistent throughout the time-lapse. Once the regions have been linked, unique IDs can be assigned to each cell. These cell IDs are distinct from cell region numbers. IDs persist throughout the entire time-lapse experiment.

We provide an alternative version of OmniSegger supporting an updated linking algorithm which is more robust to tracking cells, including those with unusual morphologies. This version of OmniSegger facilitates the use of external cell-tracking packages as well as the hand-correction of linking. The details are discussed in Appendix B2 in S1 Text.

*Cytometry.* Once the cell IDs have been assigned, OmniSegger performs cell cytometry by computing a user-specified set of cell properties, including cell length, area, birth and division (death) times, mother and daughter IDs, fluorescence intensities, fluorescent-focus position, etc. The current package implements more than 80 default cell characteristics. This cytometry step is the backbone of many analyses, and its accuracy depends sensitively on the precision of preceding steps in the pipeline.

*Data output.* A critical feature of an analysis pipeline for many users is the output of the data in a readable and easily-manipulated format. We provide several data output types that slice time-lapse data in different ways. For instance, we provide data organized by time-point (Frame files), by cell (Cell files), and holistic experimental summary (a clist file), each of which are suited to different analytical tasks. To increase the usability of the pipeline for users not familiar with MATLAB, we now output the clist file in Excel format (in addition to a mat file). This file contains >80 descriptors on a per-cell basis and >20 on a per-cell per-time-point basis, and a user can easily include more descriptors if desired.

*Data visualization.* To accompany the pipeline, we provide a data-exploration GUI which itself can be used to launch more specialized visualization tools, designed to analyze a wide range of phenomena from localization dynamics in a single cell to comparing population

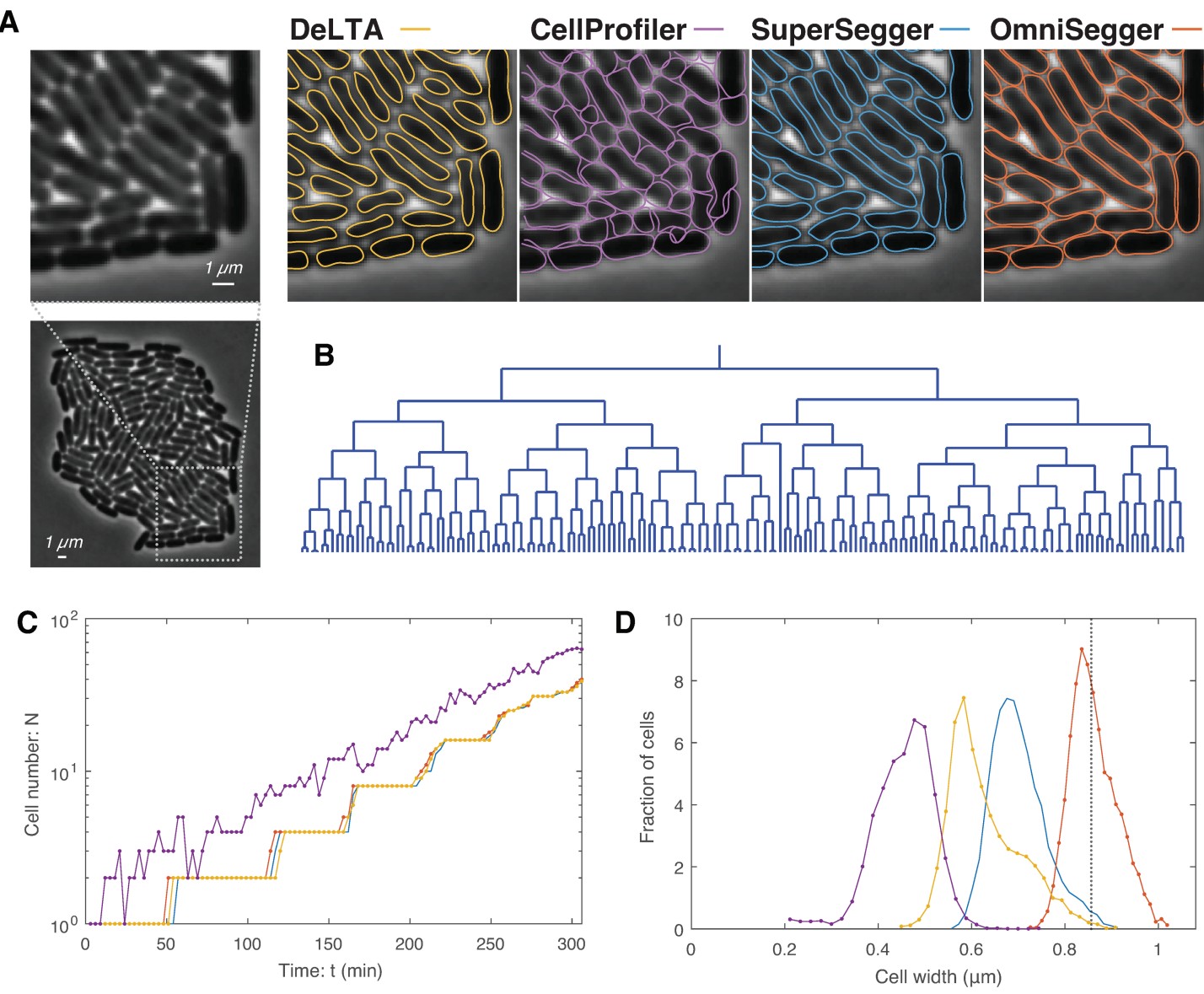

**Fig 2. Proliferation challenge.** To characterize the performances of the competing packages, we first analyzed performance under the best case scenario: the proliferation of cells, with normal morphology, from a single cell to a monolayered microcolony under ideal imaging conditions. **Panel A: Competing pipelines generate distinct cellular boundaries.** The panel shows a phase contrast image of a microcolony. Competing cell segmentations are shown for a representative magnified region. All the pipelines, except CellProfiler (purple), lead to the same number of cells and are therefore acceptable for colony-scale analysis. The remaining pipelines generate significantly different cell boundaries at a sub-cellular resolution. **Panel B: Lineage tree** as generated by OmniSegger. **Panel C: Quantitation of cell number.** CellProfiler (purple) over-segments the cells to such a great extent that it roughly double counts cells. OmniSegger, SuperSegger, and DeLTA all show comparable performance. **Panel D: Sub-cellular structure.** Panel A visually illustrates the difference in the segmented cellular boundaries. To emphasize the biological significance of these differences, we generated histograms of cellular width measured by each pipeline and compared these to the true average cellular width (dotted line, inferred from cell contact). OmniSegger both generates a measurement with the smallest bias (1%) as well as the narrowest distribution ($\sigma/\mu$ = 6%).

statistics between multiple populations. Each of these tools is provided as a MATLAB function to facilitate their adaptation to different applications. One of the new features is the inclusion of improved vector-based cell boundary representations to produce more compelling print figures for publication.

*Omnipose versus OmniSegger.* We recently described the Omnipose segmentation package [3] but we are now releasing OmniSegger. Is OmniSegger better than Omnipose? In short, Omnipose is a part of the OmniSegger pipeline. Like all other parts of the pipeline, it could be used independently; however, for most analyses additional steps are required to convert the Omnipose analyses into interpretable data. OmniSegger performs these additional steps in a package that is designed for bacterial analysis.

*Additional updated features.* In addition to the changes we have described above, we have made numerous additional changes to the OmniSegger pipeline. We provide a detailed list of feature updates in Appendix A in S1 Text.

## 3. Results and discussion

**Measuring performance.** When we described the Omnipose segmentation algorithm, we reported performance using the metric of Intersection Over Union (IOU) [3]. Although the IOU metric is widely used when measuring segmentation performance, it suffers from some serious flaws that make it less meaningful for evaluating cell segmentation performance. The first problem is that it requires a ground-truth segmentation. How should this ground-truth be generated? Although in the past we have used fluorescence imaging to help construct ground-truth masks [3], in most cases a less principled approach is used in which an expert hand-annotates the data without the aid of supplemental imaging data (e.g. [9,10]). Due to the subjective nature of these expert annotations, we believe that most IOU measurements are of little real significance for measuring the relative quality of high-performance packages (e.g. [10]). The second shortcoming of the IOU metric is that it combines many different types of errors into a single number. For instance, (i) consistently underestimating the cell width and (ii) erroneously dividing a cell into two regions can both lead to the same reduction in average IOU; however, they affect downstream analysis in very different ways. For instance, erroneous division typically requires hand correction before the pipeline can link the respective regions. We will therefore use two more specific metrics of error in our analysis: *cell width* and *fatal errors*.

*Cell width.* We analyze *cell width* as a proxy for the accuracy of sub-cellular scale cell boundaries. As we discussed above, it is technically challenging to establish a rigorous high-resolution ground-truth cell boundary; however, the cell width can be inferred with high precision precision: *E. coli* bacterial cells have a uniform rod-like morphology and grow in mono-layer microcolonies with cell contact. The bacterial width can be inferred by measuring the width of multiple cells, aligned side-by-side in a microcolony. A detailed description of the protocol is given in Appendix C in S1 Text.

*Fatal errors.* In contrast to cell width, the fatal errors rate is a measure of the cellular-scale performance of the pipeline. We define *fatal errors* as over- or under-segmentation or linking errors that require hand-correction before datasets can be temporally linked with accuracy. We will count the cumulative fatal errors during the analysis of the time-lapse dataset as a function of time. This metric has a practical interpretation: this represents the number of corrections that must be made by hand for a quantitative cell proliferation analysis. It is important to emphasize that these categories are not inclusive of all segmentation errors. For instance, in a time-lapse analysis, if a pipeline detects a cell division a frame early, as long as the regions can be correctly linked, we do not count this early over-segmentation as an error.

Cell tracking itself can present unique challenges in some contexts: For instance, if the frame rate is too low, there may be no cell mask overlap between successive frames, significantly complicating the cell tracking assignments. We discuss strategies to counteract these errors in Appendix A2 in S1 Text.

**Competing pipelines.** We will characterize the new OmniSegger pipeline, and three competing pipelines: DeLTA [11], Ilastik-CellProfiler [12,13], and SuperSegger [4] in each challenge. The first two pipelines constitute the only complete analysis pipelines that focus on single cell cytometry, require minimal user input, and are actively maintained. Though no longer maintained, we include SuperSegger to demonstrate the improvements of OmniSegger. Although packages such as FAST, CellShape, Oufti, and MicrobeJ exist for time-lapse analysis, these competing packages were not included in the comparison because their segmentation methods are limited or require significant parameter tuning for each dataset [14–17]. The features of the packages are summarized in Table 1.

**Proliferation challenge.** First, we will establish the performance of the pipelines on ideal phase-contrast time-lapse data. This performance will represent an upper bound on the performance.

*Description of the challenge.* We imaged *E. coli* proliferating from a single cell to a microcolony containing more than 100 cells. (See Fig 2A.) This data was not used to train the Omnipose phase-contrast model. It is important to emphasize that the proliferation challenge is highly non-trivial in two respects: (i) Nearly all algorithms perform well when there is minimal cell contact; however, once the microcolony grows to tens of cells in size, the shade-off phase-contrast artifact leads to reduced cell contrast in the middle of the colony. With this limitation in mind, we were careful to select tightly cropped data where the cells remain tightly focused throughout the time-lapse experiment, and we stop analysis immediately before the microcolony becomes multilayered. To prolong this period for as long as possible, we grew the cells on a 4% agarose pad as previously described [4]. (ii) The area of the colony is expanding exponentially, and therefore the speed of the edges of the colony grow exponentially as well. Tracking cells for time-lapse linking (Fig 1A) is

**Table 1. Analysis pipelines & features. A comparison of features and functions for cellular image analysis software packages.**

| Pipeline | Tuning free | Segmentation algorithm | Tracking algorithm | Data visualization | Output format | Coding Language | OS support |
|---|---|---|---|---|---|---|---|
| OmniSegger | ✓ | Deep learning (Omnipose) | Traditional | ✓ | mat, xls | MATLAB & Python | Linux, Windows, MacOS |
| SuperSegger [4] | ✓ | ML-informed Threshold | Traditional | ✓ | mat | MATLAB | Linux, Windows, MacOS |
| DeLTA [11] | ✓ | Deep learning | Deep learning | ✗ | nc | Python | Linux, Windows |
| Ilastik-CellProfiler [12,13] | ✗ | ML-informed Threshold or Watershed | Traditional | ✗ | xls | Standalone | Linux, Windows, MacOS |
| FAST [14] | ✗ | Threshold | Unsupervised learning | ✓ | mat | MATLAB, Standalone | Linux, Windows, MacOS |
| CellShape [15] | ✗ | Threshold | N/A | ✓ | N/A | Python | Linux, Windows, MacOS |
| Oufti [16] | ✗ | Threshold | Traditional | ✓ | mat, out, csv | MATLAB | Linux, Windows |
| MicrobeJ [17] | ✓ | Threshold | Traditional | ✓ | res, csv | Java (ImageJ) | Linux, Windows, MacOS |

complicated by this rapid movement of cells at the boundary of the colony at late times. With this limitation in mind, we imaged the micro-colony every 3 minutes to facilitate cell tracking late in the experiment. (The raw images and OmniSegger masks are available on Zenodo [18].)

*Pipeline performance.* OmniSegger, SuperSegger, and DeLTA all process the data without fatal errors. (See Fig 2A.) CellProfiler, which uses a threshold-based segmentation algorithm on Ilastik probability maps, has such poor performance as to make automated analysis of this dataset intractable: The pipeline overestimates the number of cells by about a multiple of two throughout the time-lapse (Fig 2C).

Except for CellProfiler, the other three pipelines process the data without fatal errors, however, there are significant differences in their performance at a sub-cellular scale. Fig 2D shows a quantitation of cell width measured by each pipeline compared with the true cell width as inferred from cell contact in the microcolony. (See Appendix C in S1 Text.) We summarize this performance in Table 2. OmniSegger shows the smallest bias and cell-to-cell variation in cell width. The differences between the package performances are significant, and we therefore conclude that OmniSegger is the only package suitable for analyses where sub-cellular scale resolution is important.

**Morphology challenge.** The segmentation of bacterial cells with unusual morphologies was a critical weakness of our original SuperSegger pipeline and a central motivation for the current work. We have encountered many experimental scenarios in our own projects where the pipeline failed, including our attempt to characterize essential-gene deletions in the model bacterium *Acinetobacter baylyi* [19] where the performance of the SuperSegger pipeline was so poor that we were forced to abandon the original aim of our experiments until we developed the OmniSegger package [6].

*Description of the challenge.* To measure the performance of competing pipelines on unusual cell morphologies, we analyze time-lapse images of cells forming filaments which traditional pipelines typically oversegment. Therefore, we collected time-lapse data from *E. coli* cells treated with a sub-Minimum Inhibitory Concentration (sub-MIC) of 10μM hydroxyurea [20,21]. Hydroxyurea inhibits DNA synthesis, and results in a phenotype of cell filamentation when below the MIC. See Fig 3A. This data was not used to train the Omnipose phase-contrast model. (The raw images and OmniSegger masks are available on Zenodo [18].)

*Pipeline performance.* The analysis of unusual cellular morphologies led to a wide range of pipeline performance. The segmentation of a representative frame is shown in Fig 3B. A schematic definition of fatal errors is shown in Fig 3C and a plot of cumulative fatal errors as a function of time in shown in Fig 3D.

CellProfiler had the worst performance. The package leads to both chronic over- and under-segmentation and generates a completely intractable number of fatal errors. Although

**Table 2. Precision in determining cell width.** We use cell width as a proxy for sub-cellular scale segmentation resolution. Comparison of bias and standard deviation in determining cell width are shown for each pipeline. Only OmniSegger generates cell boundaries precise enough for applications which depend on precise cell boundary position. OmniSegger also measures cell width with the smallest cell-to-cell variation.

| | Bias: $\Delta\mu$ | | Std: $\sigma$ | |
|---|---|---|---|---|
| **Pipeline:** | **(%)** | **(nm)** | **(%)** | **(nm)** |
| OmniSegger | 0.6% | 5 | 6% | 50 |
| SuperSegger | -18% | -160 | 7% | 60 |
| DeLTA | -27% | -230 | 9% | 80 |
| Ilastik-CellProfiler | -47% | -400 | 8% | 70 |

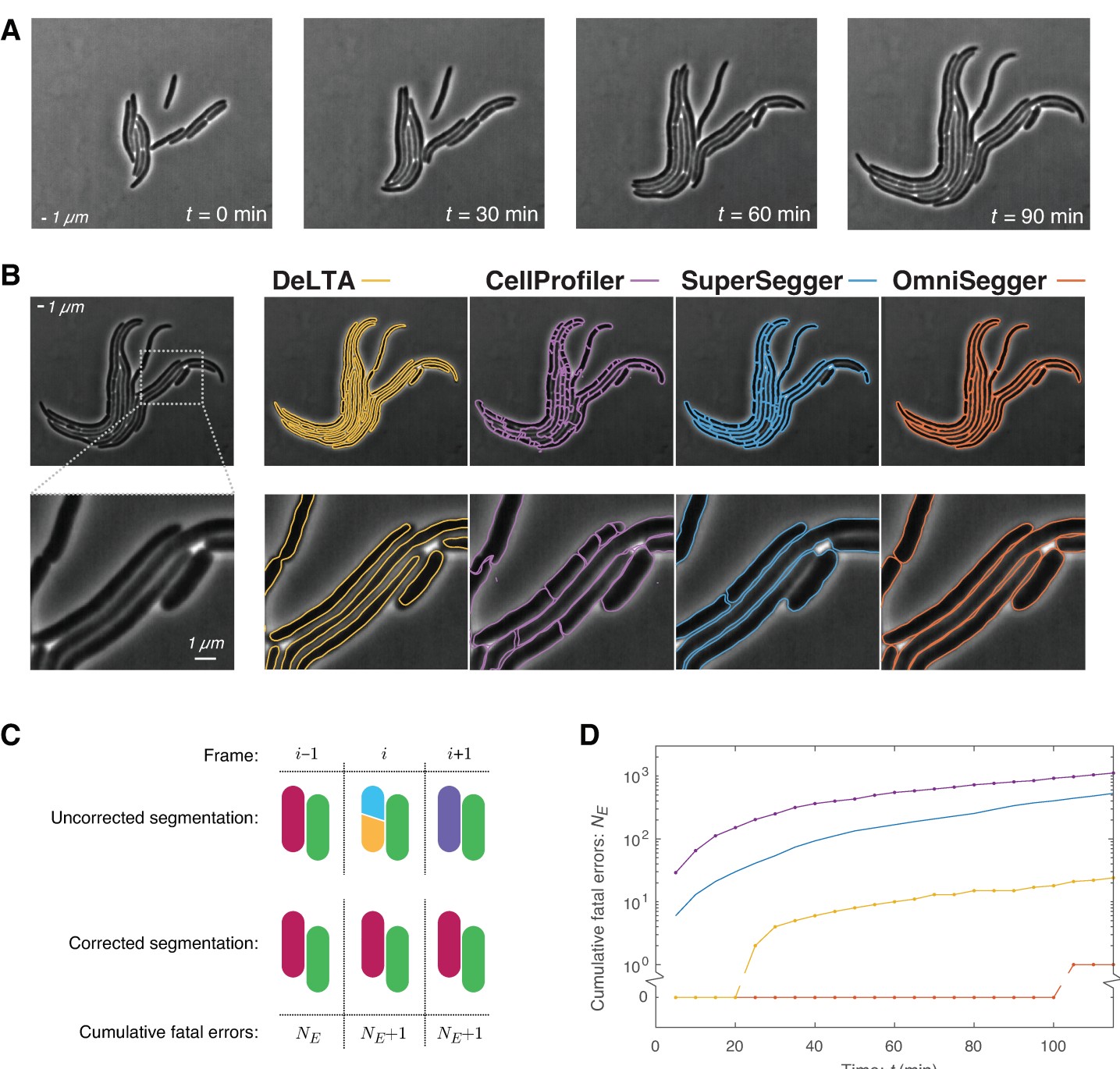

**Fig 3. The performance of competing pipelines on unusual cell morphologies. Panel A: Visualization of proliferation.** Frames from a phase-contrast time-lapse of a growing wild-type *E. coli* colony treated with a sub-MIC of hydroxyurea. Hydroxyurea inhibits DNA synthesis and results in a phenotype of cell filamentation. **Panel B: Competing pipelines generate distinct cellular boundaries.** Pipeline performance varies greatly for cells with unusual morphologies, not only at sub-cellular resolution, but at a colony scale. For this challenge, we focus on *fatal errors* defined as those that affect the cell number and prevent temporal linking without hand correction. **Panel C: Cumulative fatal error defined.** We measure performance as cumulative number of fatal errors. The red cell is over segmented in the *i*th frame, generating two new cells (yellow and cyan) before fusing back into the original cell (purple). The corrected segmentation is shown below. This segmentation error in frame *i* increases the cumulative error $N$ by 1. Note that neither erroneously narrow cell boundaries nor a late (or early) call of a cell division event constitutes a fatal error. **Panel D: Performance of competing packages measured by cumulative fatal errors.** The OmniSegger analysis is error free for 100 min of imaging (20 frames). DeLTA also results in a tractable analysis, although the analysis requires the correction of over 20 cells in a single microcolony. The performance of the SuperSegger and CellProfiler pipelines are so poor for cells of unusual morphology as to make these analyses intractable.

SuperSegger performed well on normal cell morphologies, it had extremely poor performance for filaments and generated an intractable number of fatal errors. DeLTA again fails to accurately capture cell width; however, it performs well with respect to fatal errors. For the whole dataset, only 23 fatal errors would need to be fixed, which we would define as tractable. The performance of OmniSegger is, by both performance metrics, the best. The cell boundaries are accurate at a sub-cellular scale (Fig 3B) and there is only one fatal error in the entire time-lapse which occurs very late in the experiment (Fig 3D). We therefore conclude that OmniSegger has the best performance in the context of unusual cell morphologies.

**Modality challenge.** The second focus of OmniSegger development was to add support for the segmentation of more imaging modalities. Although we prefer to use the phase-contrast modality in most experiments, some experimental scenarios demand the use of other imaging modalities. For instance, phase-contrast objectives contain a neutral-density ring that decreases the overall brightness of the objective in fluorescence applications, making it a sub-optimal modality for experiments that demand single-molecule sensitivity. We therefore worked to create a pipeline with robust segmentation performance using any of a range of canonical imaging modalities. Due to our own laboratory priorities, we introduced support for the following modalities: Phase-contrast, Brightfield, cytoplasmic fluorescence, and membrane fluorescence.

*Description of the challenge.* For this challenge, we collected data using four imaging modalities (Phase-Contrast, Brightfield, cytoplasmic and membrane fluorescence). In each case, we collected time-lapse data of *E. coli* cells proliferating over multiple generations. *For the phase-contrast modality*, we used the dataset as described in the Proliferation challenge. *For the brightfield modality*, we imaged cells using a z-stack, but focused on in-focus and under- and over-focused ($\pm$ 0.5 μm) images. We also experimented with two growth conditions: rich (LB) and minimal (M9). We did find significant differences in brightfield images and segmentation performance between these conditions and focused on the minimal media data for which the segmentation performance was highest. This dataset was not included in the ground-truth used to train the Omnipose Brightfield model. *For the fluorescence modality*, the performance is clearly dependent on the brightness of the fluorescent label. We assumed that most users would desire to use plasmid-based fluorescent fusions for this purpose. We therefore selected two representative plasmid-based IPTG inducible fusions from the ASKA collection: One with diffuse cytoplasmic localization and the other with membrane localization [5]. This dataset was not used to train the Omnipose fluorescence model. (The raw images and OmniSegger masks are available on Zenodo [18].)

*Pipeline performance.* While segmentation algorithms and pre-trained models exist to segment various imaging modalities, among the currently existing quantitative time-lapse analysis pipelines, the majority only support segmentation for phase-contrast images. In addition, alternative pipelines, which are able to segment brightfield or fluorescence images, require significant time investment to tune user-defined training or parameters  (see Table 3). OmniSegger is the only pipeline which can handle various imaging modalities out-of-the-box. As shown in Fig 4, the OmniSegger pipeline can handle all five modalities. Since other packages either do not support these alternative modalities (DeLTA, SuperSegger) or have extremely poor performance (CellProfiler), we show only the OmniSegger results. In order to evaluate the relative performance with each modality, we again quantified the cumulative number of fatal errors for cells propagating from single cells to form microcolonies (Fig 5).

For the challenge, we chose datasets that were tractable and led to high segmentation performance. In this context, all modality data were processed without fatal errors until the

**Table 3. Pipeline modality support.** For phase-contrast, **P** denotes agarose pads and **M** denotes mother machine. For brightfield, **O** denotes over-focused and **U** denotes under-focused. For fluorescence, **C** denotes cytoplasmic and **M** denotes membrane labels. * denotes tuning required.

| | Modalities: | | |
| --- | --- | --- | --- |
| **Pipeline:** | **Phase** | **Brightfield** | **Fluor** |
| OmniSegger | P | O,U | C,M |
| SuperSegger | P | ✗ | ✗ |
| DeLTA | P,M | ✗ | ✗ |
| Ilastik-CellProfiler | P* | O*,U* | C*,M* |

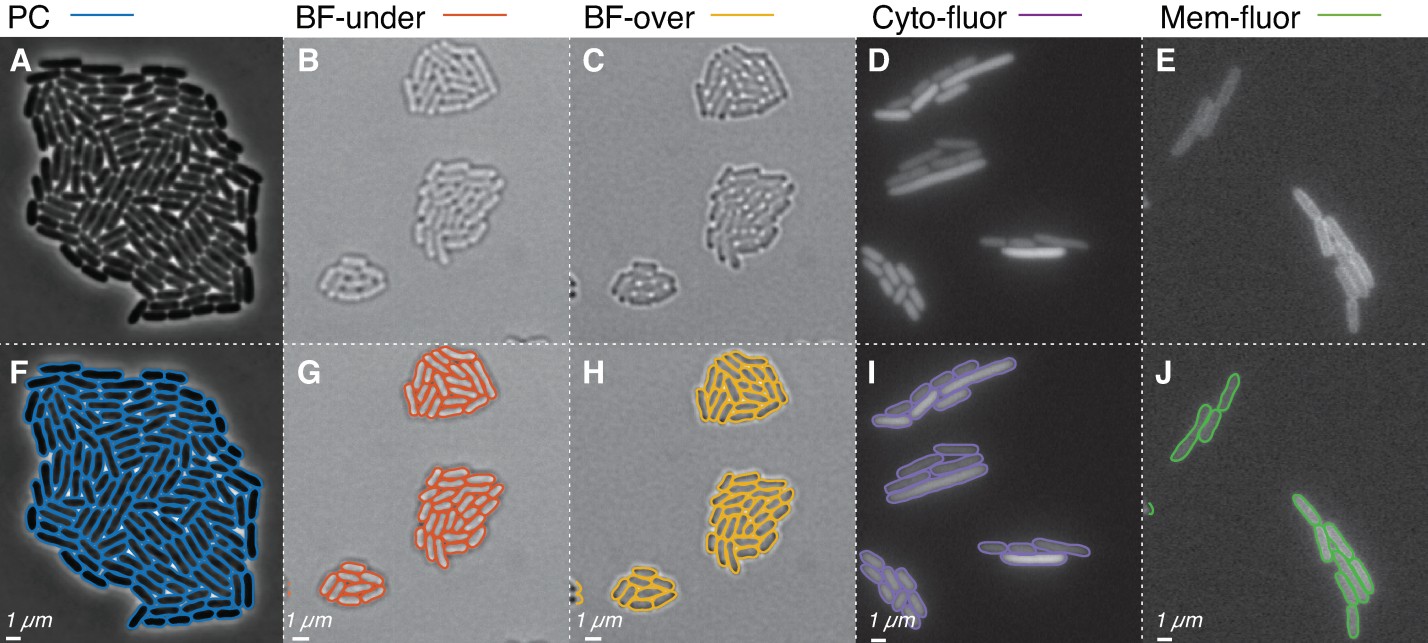

**Fig 4. OmniSegger robustly segments multiple imaging modalities. Panel A & F: Phase-contrast** (PC) imaging results in high-contrast images with well-defined cell boundaries. We find this modality generates the highest performance segmentation. **Panels B, C, G, & H: Under-focused brightfield** (BF-under) and **Over-focused brightfield** (BF-over) both generate low-contrast, noisy images. After training on this modality, we found the performance of OmniSegger to be high enough for many applications, although PC is still superior for high cell density. We also analyzed in-focus brightfield images; however, performance was not high enough to recommend this modality. **Panel D & I: Cytoplasmic fluorescence** labeling (Cyto-fluor) results in high-contrast but noisy images with bright cell interiors. We found that the Cyto-fluor modality could lead to excellent performance; however, this modality is subject to photobleaching and phototoxicity which limits the signal-to-noise ratio in some applications. **Panel E & J: Membrane fluorescence** labeling (Mem-fluor) generates high-contrast but noisy images with bright cell boundaries with nearly uniform cytoplasmic labeling from the out-of-focal-plane membrane. We found this modality could lead to high-performance segmentation; however, it was particularly sensitive to noise, especially when compared with Cyto-fluor.

microcolony reached 20 cells, where Membrane-fluorescence began to fail. Cytoplasmic-fluorescence demonstrated superior performance compared to membrane-fluorescence. Furthermore, performance on both under-focused and over-focused brightfield surpassed that of cytoplasmic fluorescence. Nevertheless, phase-contrast had the fewest fatal errors among all the aforementioned modalities. It is important to emphasize that for many applications, the performance of all modalities is sufficient to extract quantifiable data. However, we note that this challenge underestimates the advantages of phase-contrast over the competing modalities, as we later discuss in the "Which modality?" section.

*Brightfield performance.* Although the performance of OmniSegger on the brightfield images is high in the challenge, it is important to discuss a number of caveats about these

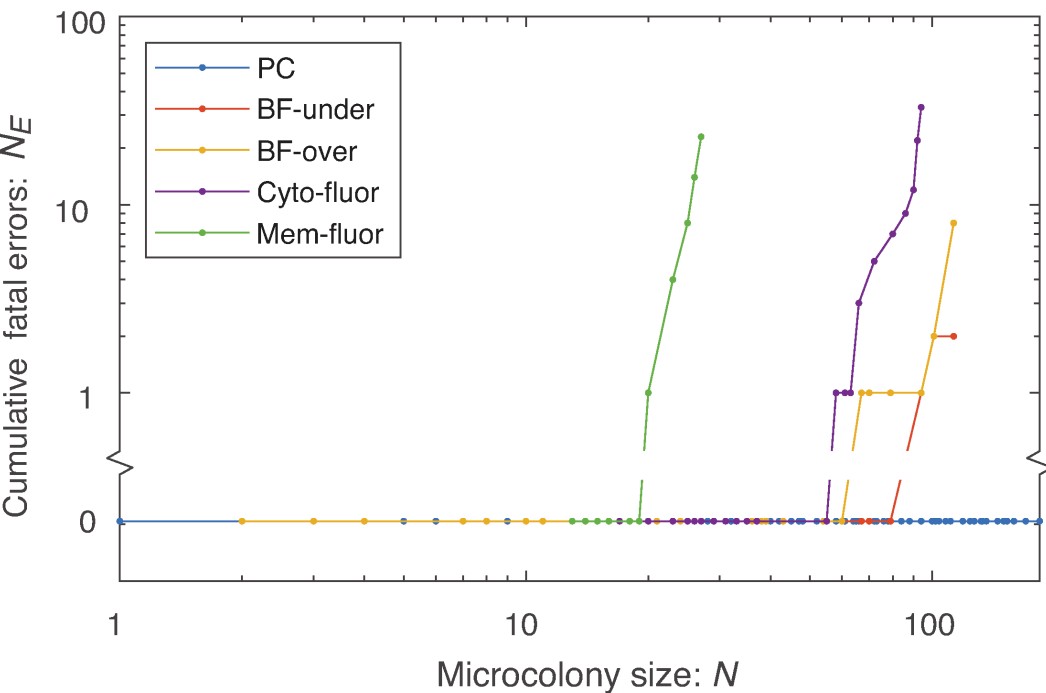

**Fig 5. OmniSegger performance for a range of image modalities.** In each case, cell proliferation was visualized on agarose pads, starting from single isolated progenitor cells growing to form microcolonies. Differences in cell growth arose from the use of distinct growth conditions and strains, therefore we quantified the fatal error number versus colony size (in cells). Phase-contrast data were analyzed without fatal errors. Over- and under-focused brightfield imaging led to high-performance segmentation as well, followed by cytoplasmic fluorescence labeling. Membrane fluorescence labeling led to the worst performance of all modalities; however, we emphasize that the performance is sufficient for many applications.

results. (i) Notably, we were unable to generate a high-performance algorithm for in-focus data in spite of extensive training. We concluded that the contrast was too low relative to the competing mechanism of contrast generation. Using the brightfield modality therefore requires a focal plane shift relative to fluorescent images, which, while tractable, is not ideal. (ii) We observed significant differences in brightfield segmentation performance depending on the growth conditions. We observed high performance on minimal media but lower performance on LB. This drop in performance was consistent with our intuition of the modality image quality when comparing the images by eye: The minimal media data was clearly more interpretable. In comparison, we find the image quality and segmentation performance for phase-contrast is much less dependent on growth conditions. We therefore conclude that the brightfield modality segmentation may have excellent performance in many contexts; however, it may be somewhat inconsistent and dependent on the dataset.

*Fluorescence performance.* There is good news for fluorescence based modalities as well: OmniSegger can segment data with very high performance. However, like the brightfield modality, there are important caveats. Clearly, both fluorescent modalities depend upon label brightness and fluorescent labels are typically subject to bleaching, which can significantly attenuate their brightness. As the intensity decreases, we found that cytoplasmic labels tend to lead to more robust performance than membrane labels. Our intuition for this difference in performance is that the contrast for the membrane label tends to be very flat across the cell, whereas the cytoplasmic label gives rise to a significant gradient towards the cell boundaries,

which is more robust to noise. (See Fig 6.) In conclusion, the fluorescence modality can lead to high performance segmentation, and the cytoplasmic label is preferable.

**Morphological robustness enables new approaches.** The development of OmniSegger enables the quantification of a variety of novel experiments involving mutants, antibiotic-treated cells, and multi-species interactions, at the single cell level. Many of these experiments involve observing a wide range of cell morphologies. Due to the challenge of accurate and robust cell segmentation, these analyses were previously not possible. The development of Omnipose enabled the segmentation step of the image analysis pipeline, and its implementation in OmniSegger introduces a powerful, highly automated tool for the biological imaging community.

**Expanded modality support offers greater flexibility.** Cell biology experiments routinely require a mix of approaches, including the use of different imaging modalities. The robust performance and analysis capabilities of OmniSegger, irrespective of the modality used, are greatly superior to other packages. Even if alternative packages could match the performance for a single modality, the generalist support offered by OmniSegger enables a uniformity in approach to multiple experiments and the advantage of avoiding package-dependent biases into an analysis.

**Understanding modality-dependent performance.** Can the observed differences in segmentation performance between modalities be rationalized based on image contrast? In Fig 6, we compare the intensity profiles from a line scan in each imaging modality. At an intuitive level, we observe that each of the modalities that is segmented with high performance (phase-contrast, under- and over-focused brightfield, and cytoplasmic fluorescence) all have consistent morphological features corresponding to each cell. In comparison, the other poor-segmentation-performance modalities (in-focus brightfield and membrane fluorescence) have

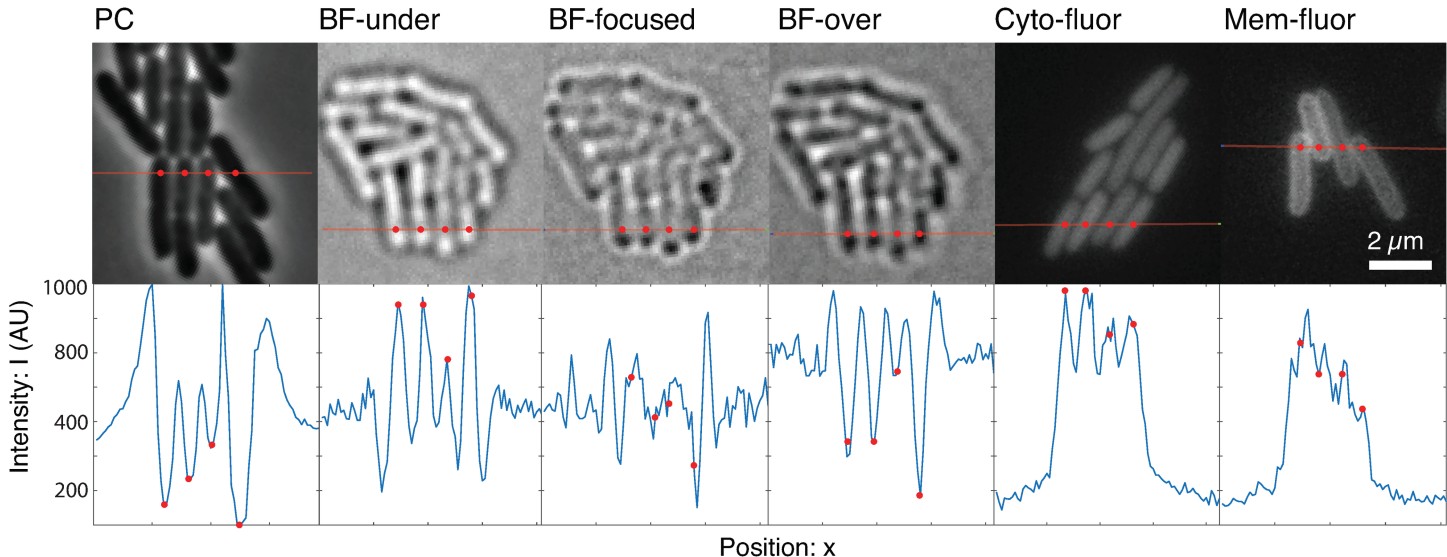

**Fig 6. Contrast in a range of modalities. Top row: Images for a range of modalities.** For each modality image, we chose a line that transects multiple cells. **Bottom row: Line scan.** The line-scan plot shows the intensity profile across the red line above. In addition, we annotated a number of points corresponding to cell centers (red points). In the modalities where the segmentation performance in highest (PC, BF-under, BF-over, Cyto-fluor), there are clear morphological features in the line-scan plot corresponding to the cell; however, in the lower performing modalities (BF-focused and Mem-fluor), the morphological features corresponding to cells are less distinct, making these modalities susceptible to contamination by noise. (Note that for visual convenience, we have applied a linear transformation on the intensities so they share a common axis.)

less consistent morphological features distinguishing neighboring cells. We therefore conclude that the relative algorithmic performance is consistent with what we would intuitively predict based on the ability of the image modality to generate unambiguous image contrast at cell boundaries.

**Which modality?** Given the flexibility to choose from multiple imaging modalities for an experiment, which modality enables the most accurate analysis? Our initial preference, based on our past work, was phase contrast imaging. Is this preference still supported in light of OmniSegger performance? Phase-contrast is, without doubt, still the best modality for determining cell boundaries for bacterial cells. In our own experience, our analysis underestimates the relative advantages of this approach. We selected datasets, growth conditions, and exposures to place the segmentation performance of each modality in the best light possible. However, in our experience, phase-contrast imaging leads to robust and reliable results with suboptimal images and it is therefore our go-to choice for imaging bacteria for quantitative analysis.

**Performance of competing packages.** We have provided extensive evidence of the performance advantages of OmniSegger over competing packages in three challenges. The machine-learning approach used in OmniSegger gives it a particular performance advantage over threshold-based pipelines [13,15–17,22,23] that are particularly sensitive to imaging conditions and morphology. Although some competing pipelines can theoretically generate comparable single-cell cytometry, including the generation of cell lineages and mean fluorescence levels [10,11,14], their poor segmentation performance limits their applicability. (It is important to emphasize that careful experimental design, that allows cells to be imaged with minimal cell contact, can significantly reduce the importance of segmentation performance in analysis. However, the development of higher-performance algorithms does allow both more cells to be analyzed as well as facilitating new types of experiments that were not previously feasible.) In other cases, packages provide high-performance segmentation but lack a complementary pipeline essential for experimental analysis [2,12,24]. A summary of features for OmniSegger and alternative packages are shown in Table 1 and more extensively in Table A in S1 Text. In summary, OmniSegger provides (i) high performance segmentation and (ii) the analysis pipeline required to convert these regions into a quantitative analysis with minimal user input or coding experience required.

## 4. Availability and future directions

The code for OmniSegger is available on GitHub: https://github.com/tlo-bot/omnisegger/.

There are still significant opportunities for the improvement of the OmniSegger pipeline for both its implementation and algorithm. *Implementation:* Currently the pipeline is implemented in a mix of MATLAB and Python code as a result of its use of both the SuperSegger and Omnipose code bases. Although some significant concerns remain about the Python language (e.g. package compatibility and failures of package maintenance), our long-term goal is to move towards a Python-only pipeline since Python is freely available, expanding the availability of our pipeline to investigators who do not have access to MATLAB. *Algorithm:* The OmniSegger pipeline does not yet make efficient use of time-persistent information in the context of segmentation. For instance, during the segmentation step, a persistent feature may be ambiguous in a single frame, but clearly resolved in the previous and subsequent frames. This issue becomes particularly problematic at high frame rates where cell division events are ambiguous over multiple frames, leading to conflicting division calls (e.g. 1

cell→2 cells→1 cell→2 cells). The pipeline does not yet allow for information to be communicated temporally between frames during the segmentation step of the analysis. We hope to introduce an algorithm that shares information between frames in the near future.

## Supporting information

**S1 Text. Supplementary details and discussion including Table A and the Materials and Methods used to generate the findings in this study.**
(PDF)

**S1 Data. Bacterial strains and ground-truth annotation for the Omnipose Brightfield model.**
(XLSX)

## Acknowledgments

The authors would like to thank S. Yang, B. Traxler, S. van Teeffelen, and J. Mäkelä.

## Author contributions

**Conceptualization:** Teresa W. Lo, Kevin J. Cutler, Paul A. Wiggins.

**Data curation:** Teresa W. Lo, Kevin J. Cutler, Paul A. Wiggins.

**Formal analysis:** Teresa W. Lo, Paul A. Wiggins.

**Funding acquisition:** Paul A. Wiggins.

**Investigation:** Teresa W. Lo, Paul A. Wiggins.

**Methodology:** Teresa W. Lo, Kevin J. Cutler, Paul A. Wiggins.

**Project administration:** Teresa W. Lo, Paul A. Wiggins.

**Resources:** Teresa W. Lo, Kevin J. Cutler, Paul A. Wiggins.

**Software:** Teresa W. Lo, Kevin J. Cutler, Paul A. Wiggins.

**Supervision:** Teresa W. Lo, Paul A. Wiggins.

**Validation:** Teresa W. Lo, Kevin J. Cutler, H. James Choi, Paul A. Wiggins.

**Visualization:** Teresa W. Lo, Paul A. Wiggins.

**Writing – original draft:** Teresa W. Lo, Paul A. Wiggins.

**Writing – review & editing:** Teresa W. Lo, Kevin J. Cutler, H. James Choi, Paul A. Wiggins.

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
