## [Decision Letter · Decision Letter 0]

13 Jan 2025

PCOMPBIOL-D-24-02059

OmniSegger: A time-lapse image analysis pipeline for bacterial cells

PLOS Computational Biology

Dear Dr. Lo,

Thank you for submitting your manuscript to PLOS Computational Biology. After careful consideration, we feel that it has merit but does not fully meet PLOS Computational Biology's publication criteria as it currently stands. Therefore, we invite you to submit a revised version of the manuscript that addresses the points raised during the review process.

Please submit your revised manuscript within 60 days Mar 15 2025 11:59PM. If you will need more time than this to complete your revisions, please reply to this message or contact the journal office at ploscompbiol@plos.org. Please include the following items when submitting your revised manuscript:

We look forward to receiving your revised manuscript.

Kind regards,

Robert F. Murphy

Guest Editor

PLOS Computational Biology

Padmini Rangamani

Section Editor

PLOS Computational Biology

**Additional Editor Comments:**

The reviews for your manuscript are now in hand. Two of the reviewers focused on specific recommended changes to improve content and clarity. The third (reviewer 3) expresses significant concern about the level of innovation, a concern that I share. This is the most important issue to be addressed if you decide to submit a revision especially in view of the journal policy that "Enhancements to existing published methods or software will only be considered if those enhancements bring exceptional new capabilities" (https://journals.plos.org/ploscompbiol/s/journal-information). Addressing the point raised by reviewer 1 about fairness of the comparison to other methods is also important. If resubmitting, please submit a detailed response to all issues raised.

**Journal Requirements:**

3) Your manuscript is missing the following sections: Design and Implementation, and Availability and Future Directions. Please ensure that your article adheres to the standard Software article layout and order of Abstract, Introduction, Design and Implementation, Results, and Availability and Future Directions. For details on what each section should contain, see our Software article guidelines:

https://journals.plos.org/ploscompbiol/s/submission-guidelines#loc-software-submissions

5) We have noticed that you have cited Table  Tables 2 and 3 not cited in the manuscript file but there is no corresponding table in the manuscript.  Please amend your manuscript to include this table noting that tables should not be uploaded as individual files.

6) We notice that your supplementary Figures, and Tables are included in the manuscript file. Please remove them and upload them with the file type 'Supporting Information'. Please ensure that each Supporting Information file has a legend listed in the manuscript after the references list.

7) Some material included in your submission may be copyrighted. According to PLOSu2019s copyright policy, authors who use figures or other material (e.g., graphics, clipart, maps) from another author or copyright holder must demonstrate or obtain permission to publish this material under the Creative Commons Attribution 4.0 International (CC BY 4.0) License used by PLOS journals. Please closely review the details of PLOSu2019s copyright requirements here: PLOS Licenses and Copyright. If you need to request permissions from a copyright holder, you may use PLOS's Copyright Content Permission form.

Potential Copyright Issues:

- Figure 1A. Please confirm whether you drew the images / clip-art within the figure panels by hand. If you did not draw the images, please provide (a) a link to the source of the images or icons and their license / terms of use; or (b) written permission from the copyright holder to publish the images or icons under our CC BY 4.0 license. Alternatively, you may replace the images with open source alternatives. See these open source resources you may use to replace images / clip-art:

8) Please ensure that the funders and grant numbers match between the Financial Disclosure field and the Funding Information tab in your submission form. Note that the funders must be provided in the same order in both places as well.

**Reviewers' comments:**

Reviewer's Responses to Questions

**Comments to the Authors:**

Reviewer #1: This paper introduces a pipeline of segmenting and analysing time lapses of growing bacterial micro-colonies (OmniSegger), which combines and improves tools of segmentation (OmniPose) and lineage tracking (SuperSegger) from the same authors. The pipeline allows the extraction of single-cell properties as a function of time including dimensions, intracellular localization of proteins, and other features. Segmentation works with different imaging modalities (phase contrast, bright field, and fluorescence), with qualitatively differences depending on contrast and noise. According to a test against two powerful alternative methods (Delta, Cellprofiler), the authors show superior segmentation and lineage tracking based on their own experimental data. The paper is well written and the tool is well described on github. This method should be very useful to a broad community of researchers studying bacteria at the single-cell level. However, I have a few comments and questions, notably regarding the comparison with alternative methods:

Comparison between methods: Did the authors train Omnipose on the same type of data used for the segmentation challenge (i.e., their own data), while the other two methods from different labs (Delta, Cellprofiler) were not retrained? It would then not be surprising that the other two methods perform less accurately. In any case, the authors should add information on training and possibly compare results from their data with results on data from other labs (like the data from the Ingalls lab (ref 21)).

Can the authors explain the results recently published by Ingalls (their ref 21), who find Delta more accurate than Omnipose for similar phase-contrast images? See my previous point.

Accordingly, the sentence ‘Although some competing pipelines can theoretically generate comparable single-cell cytometry, … [9, 12, 21], their poor segmentation performance limits their applicability.’ and similar statements throughout the results section appear too strong right now.

Further comments:

SI p4: Bactrack allows the option to use one of three different MIP solvers . What are MIP solvers? Can the authors add a sentence or two?

I didn’t quite understand whether OmniSegger uses by default a segmentation error correction for accurate lineage tracking, or if this is an add-on you actually don’t recommend using. The main text does not mention the error correction, and the SI is not very clear on which method is used. Are the data presented in Fig. 2C generated with error correction? Above which frame rate should a user consider error correction necessary, and which method would you then recommend, or do you simply not have a reliable solution for high frame rates?

Reviewer #2: This is a useful description of a natural progression of a valuable suite of tools. The results are compelling, and the manuscript is well-written. In particular, the attention to non-typical morphology and to alternate imaging modalities are welcome contributions on important topics that have received relatively little attention.

Some specific comments on the presentation:

There’s no mention of runtime in the comparisons. These would be a useful addition.

In the discussion of ‘width’ and the comparison to an average in Figure 2D, you may want to add a few words to clarify how width data was collected (to save the reader of the trouble of consulting Appendix C). The description ‘inferred from cell contact’ is not very informative. In particular, the comparison in Figure 2D seems incomplete (why not compare with a distribution?) until the details in Appendix C are consulted.

Likewise, on page 8, the statement “However, we note that this challenge underestimates the advantages of phase-contrast over the competing modalities.” Seems unjustified until the reader gets to the discussion. Perhaps could add ‘as detailed in the Discussion below’

Specific wording suggestions:

Page 1: “we have attempted to make the algorithms modular…”

This wording leaves open the question of whether you were successful. Perhaps better as “we have made the algorithms modular…”

Page 3: “The original segmentation predicted binary masks for each cell (semantic segmentation);…”

Perhaps specify for clarity: “SuperSegger’s original segmentation…”

Page 3: “This file contains > 80 descriptors on a per-cell basis and > 20 on a per-cell per-time-point basis, and is easy to expand by the user.”

‘To expand’ is unclear here: I presume this means the user can incorporate additional descriptors.

Typos:

Page 1 ‘provide a*n* open-source”

Page 3: “Each of these tools is provided as *a?* MATLAB function

Reviewer #3: In this manuscript, the authors have integrated two tools they have previously developed: SuperSegger, a time-lapse movie analysis pipeline, and Omnipose, a deep learning-based segmentation tool. The combined tool, OmniSegger, leverages the increased performance of Omnipose to fully analyze microscopy movies of coliform bacteria growing on agarose pads. The authors also adapt Omnipose to new imaging modalities, namely brightfield and fluorescence imaging. They show that their software outperforms all similar tools currently available. OmniSegger allows users to fully leverage the power of Omnipose in data analysis and promises to be useful to the community. The manuscript is well written and provides extensive details about the software implementation.

However, the manuscript does not demonstrate clear contributions in terms of originality and innovation. The main performance improvements seem to stem from the enhanced segmentation accuracy of Omnipose, while enhancements over SuperSegger’s functionalities such as data visualization, length measurement, and accessibility appear incremental.

Minor Comments:

Consider moving panels B-F from Figure 1 to later sections, or creating a Supplementary Figure, to discuss Cytometry features and the data exploration GUI in more details. A table listing all cytometry features should be added to the Supplementary Material, highlighting new or modified features compared to SuperSegger. Figure 1D should be enlarged.

The authors should provide details on the computing time required for one time-lapse movie and compare this with other available tools.

It is unclear whether the tracking/linking performance of the method was evaluated, in addition to the segmentation performance metrics such as cell width and fatal errors.

From the data availability section, it is unclear if the Fluorescence Omnipose models will be made available on Zenodo.

Page 8: "However, we note that this challenge underestimates the advantages of phase-contrast over competing modalities"  Add "(see Discussion)" or remove this sentence.

In Figure 6, please include y-axis values.

**Have the authors made all data and (if applicable) computational code underlying the findings in their manuscript fully available?**

Reviewer #1: None

Reviewer #2: Yes

Reviewer #3: Yes

PLOS authors have the option to publish the peer review history of their article (what does this mean?). If published, this will include your full peer review and any attached files.

Reviewer #1: No

Reviewer #2: No

Reviewer #3: No

**Figure resubmission:**
---

## [Decision Letter · Decision Letter 1]

23 Apr 2025

Dear Dr Lo,

We are pleased to inform you that your manuscript 'OmniSegger: A time-lapse image analysis pipeline for bacterial cells' has been provisionally accepted for publication in PLOS Computational Biology.

Best regards,

Robert F. Murphy

Guest Editor

PLOS Computational Biology

Feilim Mac Gabhann

Editor-in-Chief

PLOS Computational Biology

Guest Editor comments

As you can see below, the split between the reviewers remains on whether your work meets the novelty required for acceptance.  I believe it is indeed a borderline case, and, in such cases, I prefer favoring publication so that the community can be informed and ultimately decide upon the utility and novelty of the work.

Reviewer's Responses to Questions

**Comments to the Authors:**

Reviewer #1: The authors have satisfyingly responded to all my questions.

Reviewer #2: My concerns have been addressed.

Reviewer #3: The authors have adequately addressed most of my previous comments and concerns. I agree that the integration of Omnipose and SuperSegger into OmniSegger will be valuable to the community as it provides an accessible interface for advanced segmentation and analysis.

That said, I remain somewhat reserved about the level of novelty. While I understand that the integration efforts required to develop OmniSegger and make the power of OmniPose more accessible are significant and must be recognized, the research and technical innovations still appear limited. Moreover, the current hybrid implementation in both Python and MATLAB may reduce accessibility and ultimately the impact of the software.

OmniSegger is a much-needed tool for the community, but the contributions presented here do not constitute a fundamentally novel computational biology tool. Improving the OmniPose model capabilities and consolidating the software stack or offering a fully standalone version could significantly improve accessibility and impact.

**Have the authors made all data and (if applicable) computational code underlying the findings in their manuscript fully available?**

Reviewer #1: Yes

Reviewer #2: Yes

Reviewer #3: Yes

PLOS authors have the option to publish the peer review history of their article (what does this mean?). If published, this will include your full peer review and any attached files.

Reviewer #1: No

Reviewer #2: No

Reviewer #3: No

---

## [Editor Report · Acceptance letter]

PCOMPBIOL-D-24-02059R1

OmniSegger: A time-lapse image analysis pipeline for bacterial cells

Dear Dr Lo,

I am pleased to inform you that your manuscript has been formally accepted for publication in PLOS Computational Biology. Your manuscript is now with our production department and you will be notified of the publication date in due course.

With kind regards,

Anita Estes
